# A light-fueled self-oscillator that senses force
Zixuan Deng[1], Arri Priimagi [1], Kai Li [2] ✉ & Hao Zeng [1] ✉

Light-responsive materials with intrinsic negative feedback enable self-oscillation in non-equilibrium states. Conventional systems rely on self-shadowing in bending modes but fail when shadowing is constrained. Here, we demonstrate that external mechanical forces can bypass this limitation, enabling sustained oscillations without complete shadowing. Using a vertically suspended light-responsive liquid crystal network (LCN) strip under constant irradiation, a transition from static deformation to continuous oscillation arises when a critical load is applied. This system reveals two key phenomena: (1) oscillation amplitude scales with light intensity, reaching an angular displacement of 300°—significantly surpassing bending-mode oscillators; and (2) oscillation frequency decreases with increasing load, reflecting intrinsic mechanical sensitivity. This force-assisted self-oscillation principle generalizes across diverse deformation modes, including bending, twisting, contraction, and off-axis LCN strips. By mimicking biological mechanosensation based on dissipative mechanism, our findings provide a simplified design for non-equilibrium matter capable of dynamic adaptation to mechanical loads.

Biological organisms are self-contained, adaptive systems that transcend beyond synthetic material structures showing only stimuli-responsive behaviors[1,2]. Living systems integrate intricate feedback and feedforward loops through hierarchical regulatory strategies spanning physical structures, biochemical pathways, and mechanical interactions, to maintain homeostasis while adapting to environmental changes through dynamically adjusting their behavior[3–5]. A defining characteristic of biological systems is their operation far from thermodynamic equilibrium, where energy dissipation drives non-equilibrium functions such as autonomous motion, rhythmic oscillations, and adaptive decision-making[6–8]. In these systems, external stimuli act as perturbations rather than direct commands, triggering coordinated responses that emerge from the interplay of structural, chemical, and energetic factors[9–11]. This ability to couple sensing to actuation through dissipative processes, enabling self-regulation and adaptive responses, is often referred to as *natural intelligence*[12,13]. In contrast, engineered systems—even those leveraging physical intelligence or embodied computation—remain primitive in their ability to self-regulate and autonomously adapt compared to biological counterparts[14–16].

To bridge this gap, materials scientists have developed stimuli-responsive materials capable of shape-morphing[17,18], autonomous locomotion[19–21], tactic responses[22–24], and inter-agent communication[25–27], aiming to replicate non-equilibrium behaviors observed in biological systems[28–30]. A particularly promising approach is the realization of self-oscillation[31], where materials undergo periodic motion in response to

constant stimulus without the need for complex external controls[32–34]. One widely studied mechanism is self-shadowing-induced oscillation in light-responsive soft materials[35–37]. This effect arises from a dynamic feedback loop between structural deformation and energy dissipation, where periodic bending and relaxation occur as light absorption is cyclically interrupted[38]. However, self-shadowing-based oscillators face critical constraints. They primarily rely on bending deformation, which restricts motion to small amplitudes (typically <50°)[39–41]. Additionally, the inherent softness of these materials limits their ability to withstand external forces[38,42,43], making them prone to destabilization and constraining their practical applications. Overcoming these constraints requires new strategies to enable self-oscillation in a broader range of deformation modes while enhancing mechanical robustness and functional adaptability[44–46].

Here, we present a force-assisted self-oscillation strategy that generated by twisting-induced reductions in light absorption. Our system, composed of liquid crystal network (LCN) applied to an external force field and driven by a constant illumination, achieves large oscillation amplitudes of up to 150°, significantly exceeding the limits of bending-based oscillators. This force-assisted oscillation allows programmable oscillation dynamics by modulating the external load, exhibiting force-sensitive behavior. Moreover, this mechanism remains universally applicable across various light-responsive LCNs, including all fundamental deformation modes as well as off-axis LCN strips.

[1]Faculty of Engineering and Natural Sciences, Tampere University, Tampere, Finland. [2]School of Civil Engineering, Anhui Jianzhu University, Hefei, China. ✉e-mail: kli@ahjzu.edu.cn; hao.zeng@tuni.fi

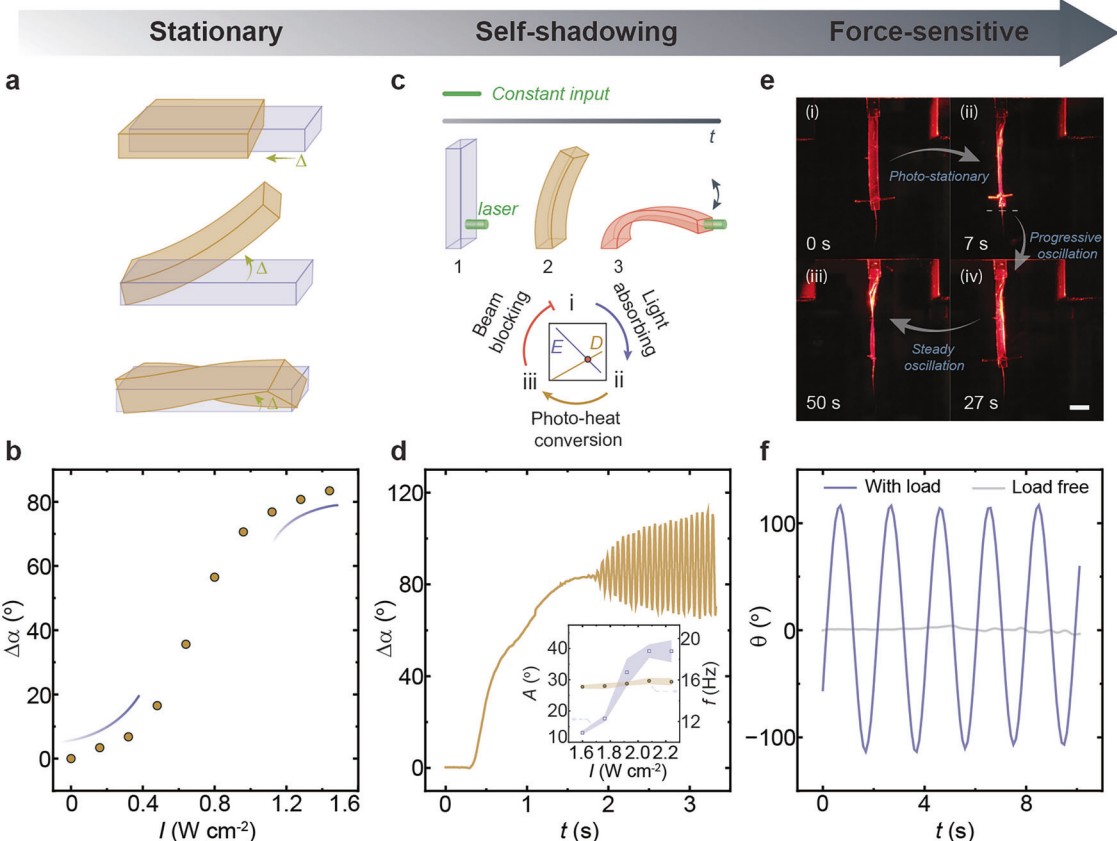

**Fig. 1 | System concept. a** Schematic representation of the three fundamental deformation modes in liquid crystal network (LCN) actuators. **b** Evolution of the deflection angle ($\Delta\alpha$) as a function of light intensity. **c** Top: sequential illustration of a planar-aligned LCN strip bending over time under continuous illumination. Bottom: flowchart depicting the self-shadowing effect, where a negative feedback loop emerges from structural deformation ($D$) and light absorption ($E$). **d** Time-resolved deflection angle of a planar-aligned LCN strip under continuous illumination. Irradiation conditions: 532 nm, 1.9 W cm$^{-2}$. Inset: relationship between oscillation amplitude and frequency as a function of light intensity. The error bars in inset are displayed as mean values +/− standard deviation ($n$ = 3). The same sample was measured repeatedly. **e** Photographic sequence showing the force-sensitive self-oscillator made from a planar-aligned LCN strip under rectangular-shaped illumination when loaded with a 1.8 g weight. Irradiation conditions: 532 nm, 0.2 W cm$^{-2}$. Scale bar: 5 mm. **f** Time-resolved oscillation profiles comparing the loaded and load-free conditions. All samples thickness: 100 μm.

## Results

When exposed to light, a photothermal polymer absorbs photons and converts them into heat, driving a shape transformation from its original configuration to a new stationary state. This process is governed by photothermal transduction, where absorbed energy is dissipated into the surrounding environment through a thermal gradient. Among such materials, thermally responsive LCNs undergo shape transformations in three fundamental deformation modes: contraction-expansion, bending-unbending, and twisting-untwisting (Fig. 1a). These deformations are dictated by the molecular alignment within the polymer matrix, a subject extensively studied over the past decades[47]. To quantify shape transformation, a shape-morphing parameter ($\Delta$) is introduced, which measures displacement, out-of-plane bending angle, or twisting angle, depending on the deformation mode. Typically, $\Delta$ follows a nonlinear response to light irradiation, as illustrated in Fig. 1b. For example, when a planar-aligned LCN strip is illuminated, the front facet contracts more than the back side, causing the strip to bend. This asymmetric contraction-induced bending increases with light intensity, reaching a maximum deflection angle ($\Delta\alpha$) of approximately 83° in the photothermal stationary state, where the light absorption area stabilizes. For 100 μm-thick samples, light-induced deformation is polarization-independent because the thickness enables near-complete absorption of incident light across all polarizations, leading to sufficient photothermal heating[41]. Further details on material properties and deflection kinetics are provided in Supplementary Figs. 1 and 2.

Achieving self-sustained oscillation requires breaking this stationary condition. One well-known approach involves using a focused light beam to confine the illuminated area, increasing the local intensity to a critical point where the self-shadowing effect emerges—a widely employed mechanism in light-driven self-oscillatory systems (Fig. 1c)[39,48]. Here, excessive bending blocks incident light, leading to localized cooling and relaxation of the material. As the material unbends, it re-exposes itself to the light source, initiating a new oscillation cycle. This phenomenon arises from a negative feedback loop between light absorption, deformation, and shadowing, which regulates the oscillatory behavior (Fig. 1d). In this scenario, the oscillation frequency is primarily determined by the system's resonant frequency and remains relatively insensitive to light power. However, the oscillation amplitude increases with higher input power, as shown in the inset of Fig. 1d. Further details on self-shadowing-induced oscillation kinetics are provided in Supplementary Fig. 3.

In contrast to self-shadowing oscillators, introducing an external force field enables a distinct oscillatory mechanism. When an LCN strip is freely suspended from a fixed platform, gravity flattens it along the vertical direction. Upon rectangular-shaped illumination, the strip deforms following the same principles as described in Fig. 1a. However, when an external force field is introduced—by attaching a hanging weight via a thread fixed to the strip's end—the material undergoes force-assisted self-oscillation, exhibiting alternating clockwise and counterclockwise twisting around the thread axis (Fig. 1e). This force-assisted mechanism allows the system to surpass its photo-stationary state, enabling oscillation under

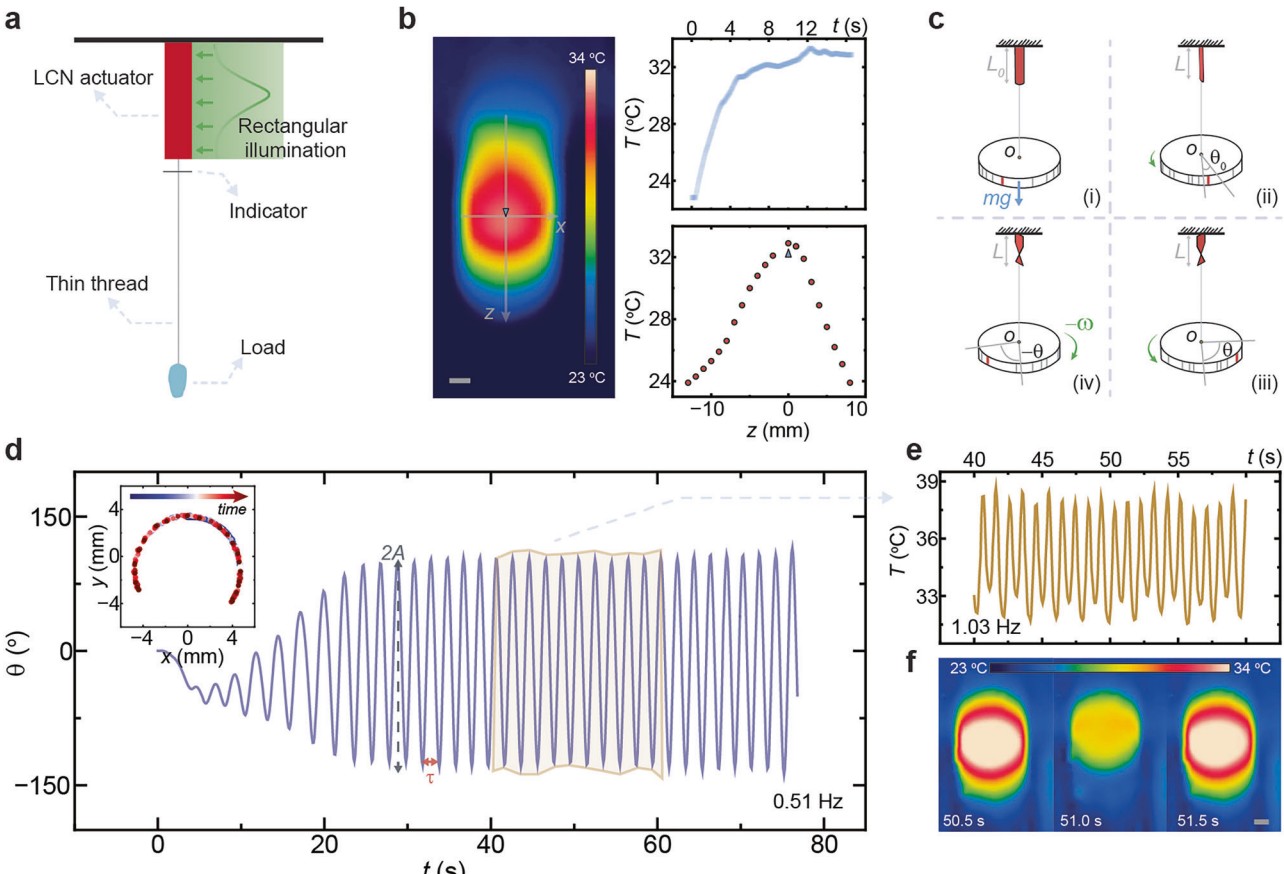

**Fig. 2 | Force-field-induced self-oscillation. a** Schematic of the force-sensitive self-oscillation device. **b** Infrared thermal imaging of an LCN strip aligned with the director axis under rectangular-shaped illumination. Left: thermal map of temperature distribution. Top right: time evolution of the temperature at the marked point during illumination. Bottom right: temperature distribution along the z-axis. Load condition: 1.8 g weight. Irradiation conditions: 532 nm, 84 mW cm$^{-2}$. **c** Mechanistic illustration of force-induced self-oscillation. **d** Time evolution of oscillatory rotation under an external load. Inset: x-y trajectory of the indicator tip during oscillation. **e** Magnified view of the temperature oscillations over several cycles. **f** Sequential photographic frames highlighting thermal fluctuations within a single oscillatory cycle. Load condition: 1.8 g weight. Irradiation conditions: 532 nm, 167 mW cm$^{-2}$. All scale bars: 2 mm.

continuous rectangular-shaped illumination without stalling (Fig. 1f), with the oscillation characteristics strongly depending on the applied load.

Figure 2a illustrates the experimental setup for force-assisted self-oscillation. An LCN strip is suspended from a fixed glass ceiling, with a weight attached via a long, thin copper thread (120 μm thick, 1.1 m long). The long copper thread minimizes the influence of the weight's moment of inertia, ensuring that the applied force remains strictly vertical. The actuator is illuminated from the front by a rectangular-shaped light beam (spot size: 30 × 4 mm$^2$), and a small filament attached to the thread serves as an indicator to track the rotation angle ($\theta$). Unlike conventional self-shadowing oscillators, where a focused light beam selectively excites a portion of the material, this setup employs large-area illumination across the entire strip. Upon photothermal heating, the actuator's temperature rises rapidly, reaching a peak of 33 °C (Fig. 2b, left). The temperature increase exhibits a Gaussian-like distribution in the x-z plane due to the laser's inherent Gaussian beam profile (Fig. 2b, bottom right) and occurs within seconds (Fig. 2b, top right). Further details on temperature kinetics under different light intensities, as well as a detailed analysis of spatial temperature distribution, are provided in Supplementary Figs. 4 and 5.

In this system, the vertical force field suppresses bending deformation, confining the actuator's response to changes in height and twist angle. The self-oscillation process unfolds as follows: Upon illumination covering the entire front surface, the material absorbs light and contracts, shortening its length from $L_0$ to $L$, thereby lifting the attached weight. Simultaneously, the strip twists by an initial rotation angle ($\theta_0$) due to inhomogeneities across its

width (Fig. 2c, i–ii). As the light energy continues to be supplied, the actuator twists further to a maximum angle $\theta$ (Fig. 2c, ii–iii). However, at larger $\theta$, the increasing incidence angle of illumination reduces the effective photothermal absorption. At this point, the active force generated by light absorption is insufficient to counteract the restoring force from the external load, triggering a reverse rotation (Fig. 2c, iii–iv). The actuator returns to the initial state and a new cycle begins, resulting in a continuous, self-sustained oscillation with a characteristic oscillation period $\tau$, and rotation angle ($\theta_{max}$) per cycle.

This behavior is captured in Fig. 2d, where the LCN initially undergoes a transient twisting phase before stabilizing into periodic oscillations. Over time, the rotation amplitude gradually increases until reaching a steady oscillation with a constant amplitude ($A = \frac{1}{2}\theta_{max}$) and $\tau$. This progression is evident in the trajectory of the indicator's tracked spot, shown in the inset of Fig. 2d. During the early phase, the oscillation trajectory is limited, but as the system stabilizes, it evolves into a well-defined periodic path. For instance, when a 1.8 g attached weight is used, the actuator oscillates at a steady frequency of 0.51 Hz. Thermal imaging analysis reveals that the actuator's temperature oscillates between 32 °C and 39 °C, but at a frequency twice that of the mechanical oscillation—1.03 Hz (Fig. 2e), indicating that the temperature peaks twice per oscillation cycle: once during clockwise twisting and once during counterclockwise twisting (Fig. 2f). Further details on temperature oscillations under varying light intensities are provided in Supplementary Fig. 6. When varying the number of connecting points by increasing the number of wires, the oscillation dynamics remain consistent

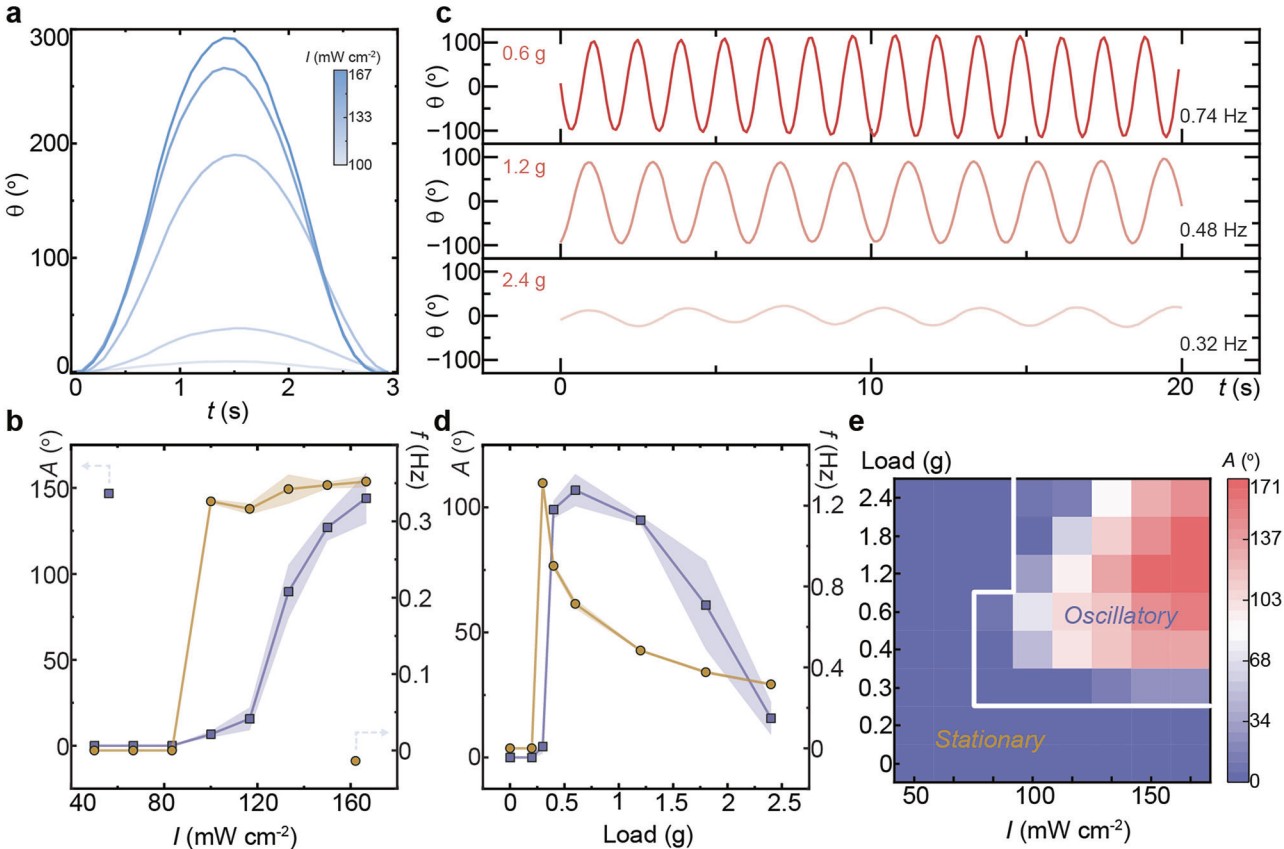

**Fig. 3 | Mechanosensation. a** Rotation angle during a single oscillation period under varying light intensities. **b** Relationship between oscillation amplitude and frequency as a function of light intensity. Load condition in (**a,b**): 2.4 g weight. **c** Stacked oscillation patterns under different loads. **d** Summary of oscillation amplitude and frequency across various load conditions. Irradiation conditions: 532 nm, 117 mW cm$^{-2}$. **e** Heat map showing the oscillation thresholds and amplitudes across a range of illumination intensities and attached loads. The error bars in (**b**), (**d**) are displayed as mean values +/− standard deviation ($n = 3$). The same sample was measured repeatedly.

under an identical external load. Only minor variations are observed in the oscillation amplitude (<10%) and frequency (<5%) when transitioning from a single-point to a five-point contact (Supplementary Fig. 7). These small differences indicate that our self-oscillation mechanism is highly robust to changes in contact area and largely insensitive to variations in stress distribution at the strip's end.

The oscillatory behavior of the system is strongly influenced by light intensity. As shown in Fig. 3a, $\theta$ increases significantly with higher light intensity, culminating up to 300°. This indicates that the system's response is highly sensitive to the amount of light energy supplied. Despite these large deformations, oscillation frequency remains relatively stable (Fig. 3b). This suggests that while light intensity governs the amplitude of deformation, the oscillation period is primarily dictated by the system's mechanical properties including the inertia and stiffness that are unaffected by light intensity. Further details on oscillation behavior under varying light intensities are provided in Supplementary Fig. 8.

Beyond irradiation, external loading plays a crucial role in shaping the oscillatory dynamics, as shown in Fig. 3c. Increasing the attached weight significantly reduces both oscillation amplitude and frequency (Fig. 3d). This inverse relationship arises because a heavier weight increases the restoring force opposing photothermally induced contraction that elevates the height of the load and the inertial effect of system, thereby limiting the extent of twisting deformation and slowing the oscillation. Further experimental details on oscillation behavior under varying loads are provided in Supplementary Fig. 9. It is worth noting that while the applied load affects the oscillatory behavior, it has a negligible impact on the material's intrinsic properties, such as Young's modulus (Supplementary Fig. 10a, b). The crosslinked polymer strips exhibit reliable reversibility and robust oscillation

dynamics under repeated loading-unloading cycles (Supplementary Fig. 10c).

The interplay between active photomechanical forces and stretching-induced resistance establishes a critical threshold for oscillation. If the applied load or light intensity falls below a certain limit, the system fails to sustain oscillation, as shown in the amplitude map of Fig. 3e. Another key factor influencing oscillatory response is the length of the LCN strip. Experimental results reveal that shortening the actuator enhances oscillation frequency, with the frequency doubling to 2 Hz when the strip length is almost halved from 22 mm to 12 mm (Supplementary Fig. 11). Interestingly, slightly above the threshold of applied load, where the competition between forces becomes sensitive, the system exhibits spontaneous switching between two distinct oscillation states—a behavior likely driven by nonlinear interactions between photomechanical forces and external loading (Supplementary Fig. 12).

The force-field-assisted self-oscillation mechanism extends to diverse director patterns along the strip's long axis, demonstrating its versatility across a wide range of light-responsive polymer strips. The distinct deformation is achieved by manipulating the orientation of liquid crystal molecules between two glass slides (Fig. 4a). Under rectangular-shaped illumination, a planar-aligned LCN strip naturally bends toward the light source due to differential contraction between its front and back surfaces, a result of photothermal induced contraction (Fig. 4b). The deformation behavior can be further tailored by cutting the strip at an off-axis angle ($\varphi$) relative to the molecular director (**n**), defined in Fig. 4a. For instance, when an LCN strip is cut at $\varphi = 45°$ and subjected to a 2.4 g load, it exhibits periodic oscillation under illumination, with an amplitude of approximately 75° and an oscillation frequency of 0.42 Hz

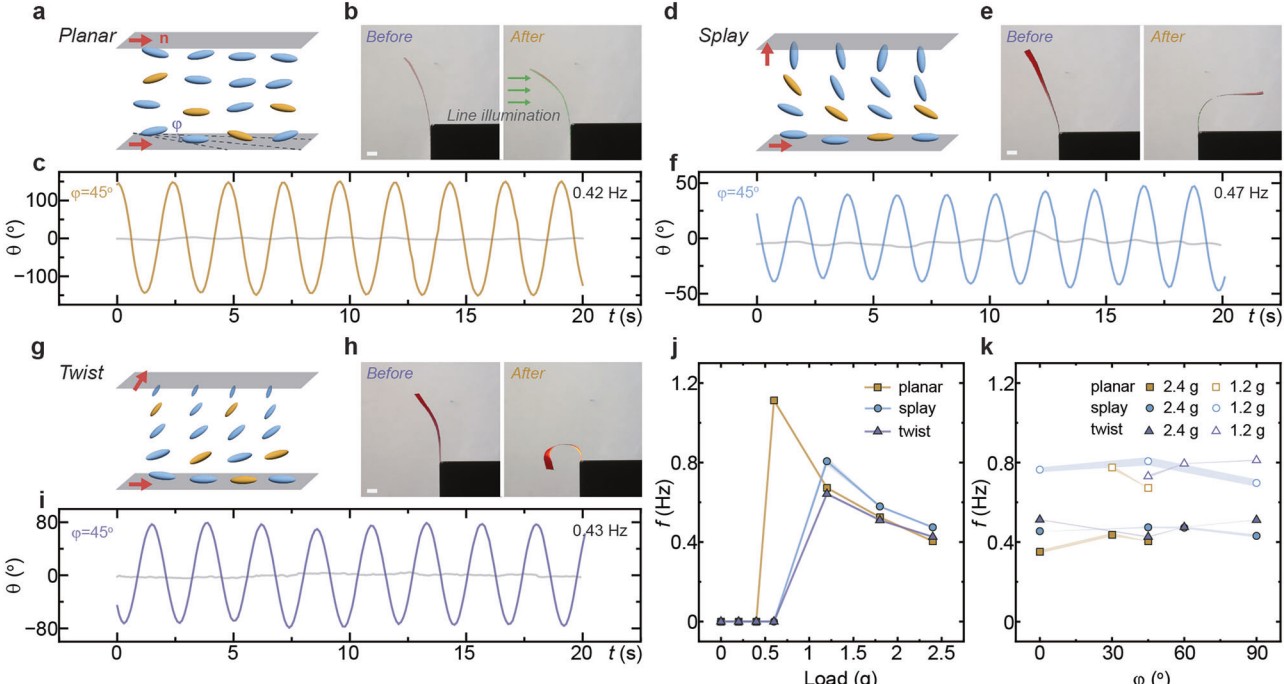

**Fig. 4 | Ubiquity of force-sensitive self-oscillation. a** Schematic of planar-aligned LCN with the off-axis angle ($\varphi$) defined as the angle between the cutting direction and molecular director **n**. **b** Photos of $\varphi = 0°$, corresponding to director along the long axis of the planar-aligned LCN strip, bending toward the light source at 117 mW cm$^{-2}$ intensity. **c** Time-evolving oscillation of $\varphi = 45°$, planar LCN under a 2.4 g load. Irradiation conditions: 532 nm, 500 mW cm$^{-2}$. **d** Schematic of splay-aligned LCN. **e** Photos of $\varphi = 0°$, splay LCN strip bending away from the light source at 117 mW cm$^{-2}$ intensity. **f** Time-evolving oscillation of $\varphi = 45°$, splay LCN under a 2.4 g load. Irradiation conditions: 532 nm, 233 mW cm$^{-2}$. **g** Schematic of twist-aligned LCN. **h** Photos of $\varphi = 0°$, twist LCN forming a spiral at 117 mW cm$^{-2}$ intensity. **i** Time-evolving oscillation of $\varphi = 45°$, twist LCN under a 2.4 g load. Irradiation conditions: 532 nm, 550 mW cm$^{-2}$. **j** Summary of load effects on oscillation frequency and (**k**) summary of off-axis angle's effect on oscillation frequency across all deformation modes. The error bars in (**j**), (**k**) are displayed as mean values $+/-$ standard deviation ($n = 3$). The same sample was measured repeatedly. All scale bars: 2 mm.

(Fig. 4c). Further details on the influence of the off-axis angle are provided in Supplementary Fig. 13.

This underlying mechanism is not limited to a single deformation mode but operates across other alignment configurations. In splay-aligned LCN strips (Fig. 4d), the structure bends away from the homeotropic surface (Fig. 4e), while twist-aligned LCN strips (Fig. 4g) form a spiral-like shape upon light exposure (Fig. 4h). These differences arise from variations in molecular orientation within the polymer network, and their shape-morphing behaviors under load-free condition are captured in Supplementary Movie 1. Similar oscillatory behavior is observed in these systems when strips are prepared with an off-axis angle. Regardless of the deformation mode, the oscillation frequency remains largely dependent on the attached weight, stabilizing to 0.4–0.5 Hz under a 2.4 g load, as shown in Fig. 4f, i. Further details on the effect of the off-axis angle for splay- and twist-aligned configurations are provided in Supplementary Figs. 14 and 15, and in Supplementary Movie 2.

A key characteristic of the force-field-assisted self-oscillation is its load-dependent behavior, which occurs across all three deformation modes. As shown in Fig. 4j, the oscillation frequency decreases as the load increases, a trend that aligns with the previously observed oscillation dynamics. Meanwhile, the oscillation frequencies of the planar, splay, and twist modes converge at loads above approximately 1 g. This convergence is likely due to the dominance of load-induced restoring torque under heavy loading, which governs the cycle time and effectively masks differences in photomechanical responses among the alignment modes. Notably, for various off-axis angles derived from these modes, the oscillation frequency depends on the load's weight but stabilizes near corresponding resonant values. For example, the frequency groups around 0.4 Hz for a 2.4 g load and 0.7 Hz for a 1.2 g load, as summarized in Fig. 4k. Further details summarizing the effect of the off-axis angle on

self-oscillation for the three deformation modes are given in Supplementary Fig. 16. This resonance-adaptive behavior highlights the versatility of the force-field-assisted self-oscillatory mechanism across diverse material configurations and mechanical responses.

## Discussion
Despite significant advances, fundamental gaps persist between bioinspired soft robots and their biological counterparts in varied aspects, such as their hierarchical architectures and dynamic functionalities. A hallmark of living systems is their ability to operate far from equilibrium, allowing for remarkable adaptability and resilience. This quality is exemplified in diverse oscillatory behaviors, including neuronal firing, heart pulsation, circadian rhythms, and the beating of flagella and cilia[49,50].

Autonomous behavior in responsive materials has been extensively studied, where periodic motions emerge spontaneously under constant energy input[33,48]. In such systems, external triggers are redefined as energy reservoirs, consumed in a dissipative manner to disrupt equilibrium states without centralized control[9]. Traditionally, self-sustained motion in photomechanical polymers has relied on the self-shadowing effect, wherein structural deformation blocks light absorption to establish a negative feedback loop. This mechanism typically necessitates localized laser illumination to maintain oscillatory motion. In contrast, the force-field-assisted strategy reported here achieves twisting-induced reductions in light absorption, enabling continuous oscillation under large-area illumination. This behavior emerges from the interplay between photogenerated forces and stretching-induced resistance, which significantly enhances the responsive delay, allowing the system to cross equilibrium and sustain oscillatory motions above a critical stretching threshold. By eliminating the need for selective light exposure, this approach significantly broadens the accessibility of oscillatory behaviors to a wider range of responsive materials,

including LCNs with distinct deformation modes and obliquely cut LCN strips.

Mechanosensation—a universal feature of living systems—is crucial for sensing and responding to mechanical cues[51,52]. It is exemplified in diverse biological structures, such as cochlear hair cells[53], spider sensilla[54], and the flagella or cilia of microorganisms[55,56], which exhibit responsiveness to external forces. The force-field-induced oscillator not only captures the far-from-equilibrium dynamics but also exhibits programmable oscillatory behavior modulated by external loads, drawing parallels to biological mechanosensation. The increase in oscillation amplitude with increasing light intensity mirrors the behavior of flagella beating, which is highly dependent on ATP hydrolysis. Inner arm dyneins—motor proteins driving flagellar motion—exhibit greater ATP affinity at elevated concentrations, resulting in increased wave amplitudes[57]. Similarly, in the force-field-assisted system, greater energy input enhances oscillation amplitude, reflecting its dependence on available energy reservoir. Furthermore, biological flagella and cilia demonstrate adaptive responses to environmental mechanical cues, such as changes in medium viscosity. Increased hydrodynamic resistance opposes dynein-generated forces, reducing flagellar beating amplitude and, in extreme cases, leading to stalling[58]. Comparable dynamics are evident in the force-field-assisted oscillator, where greater external loads suppress oscillation amplitude by counteracting photomechanical actuation. Flagellar oscillation frequency is robust to external noise, such as turbulent fluid flow, due to intricate biochemical feedback mechanisms within the axoneme[59]. While the force-field-assisted oscillator lacks such complex regulatory pathways, it demonstrates notable frequency stability across different material configurations and alignments. However, unlike flagella, its frequency is more susceptible to low-to-moderate external loads. Future research may incorporate active feedback circuits to dynamically adjust light intensity in response to load changes, enabling consistent oscillation frequency and closer mimicry of biological mechanosensation. The force-sensing capability of the system is further validated through a quantitative demonstration: by using oscillation frequency as the readout parameter, unknown applied forces can be reliably inferred from a calibration curve. As shown in Supplementary Fig. 17, the predicted weights exhibit strong agreement with the actual values, with relative errors below 8%. This result marks an important first step toward developing an active, light-driven soft force sensor.

In summary, this work presents a simple approach to achieving self-sustained oscillations in light-responsive soft materials by leveraging external force fields. Using a hung liquid crystal network strip as a model system, we demonstrate that the interaction between photogenerated forces and stretching-induced resistance can drive continuous oscillations. The oscillation frequency is predominantly governed by the system's mechanical properties, while the amplitude increases with higher energy input, reaching rotation angles of up to 300°, significantly surpassing those observed in bending modes. Under varying external loads, both the oscillation frequency and amplitude exhibit distinct force-dependent responses, displaying force-sensitive behaviors that echo the dynamics of biological systems. This force-field-assisted self-oscillation phenomenon is observed across a wide range of light-responsive materials, including LCNs in all fundamental deformation modes and off-axis LCN strips. Notably, frequency stability is maintained across different configurations and material alignments. By drawing analogies to the mechanosensation observed in biological oscillators, our findings present a novel approach to designing bioinspired systems with enhanced autonomy and adaptability.

## Methods

### Materials in brief

1,4-Bis[4-(3-acryloyloxypropyloxy)benzoyloxy]-2-methylbenzene (97%, ST03021) and 4-Methoxybenzoic acid 4-(6-acryloyloxy-hexyloxy)phenyl ester (97%, ST03866) were obtained from SYNTHON Chemicals. 2,2-Dimethoxy-2-phenylacetophenone was obtained from Sigma-Aldrich. Disperse Red 1 was obtained from Merck. Copper wires were obtained from a cell phone charging cable. All chemicals were used as received.

### Sample fabrication

Liquid crystal cells were prepared by gluing two functionalized glass substrates, either with rubbed polyvinyl alcohol (PVA, 5 wt% water solution, 4000 RPM, 1 min, baked at 100 °C for 10 min) or homeotropic command layer (JSR OPTMER, 6000 RPM 1 min, first baked at 100 °C for 10 min, followed by 180 °C for 30 min) for tailored alignment. 100 μm microspheres (Thermo scientific) were used as spacers to determine the film thickness. The liquid crystal mixture was prepared by mixing 77 mol % ST03866, 20 mol% ST03021, 2 mol% Disperse Red 1, and 1 mol% of 2,2-Dimethoxy-2-phenylacetophenone on a magnetic stirrer at 70 °C (400 RPM) for 30 min. Then the mixture was filled into the cell via capillary force at 70 °C and maintained for 10 min before cooling down to 30 °C (5 °C min$^{-1}$). The sample was irradiated with UV light (365 nm, 180 mW cm$^{-2}$, 10 min) for polymerization. Finally, the cell was opened, and LCN strips were cut out from the sample film with specific off-axis angles by using a blade.

### Optical characterization

Photographs and movies were captured with Canon 5D Mark III camera with 100 mm lens. Thermal images were recorded with an infrared camera (FLIR T420BX) with a close-up 2x lens. A continuous-wave laser (532 nm, 2 W, ROITHNER) was used for light excitation, and the rectangular-shaped laser beam (spot size: 30 × 4 mm$^2$) was generated by passing a cylindrical lens.

### Data analysis

The movement was recorded, and quantitative data were extracted from the movie with a video analysis software (Tracker).

### Data availability

The data that support the findings of this study are available from the corresponding authors upon request. The raw data generated in this study have been deposited in Fairdata QVAIN online storage space at https://doi.org/10.23729/fd-7e0e4ec6-4b65-31de-bdeb-50571093deb2.

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

## Acknowledgements

We gratefully acknowledge funding from Academy of Finland (Research Fellowship, No. 340263, to H.Z., Center of Excellence in Life-Inspired Hybrid Materials, No. 346107 and the Flagship Programme on Photonics Research and Innovation No. 320165, to A.P.). Z.D. is supported by European Union's Horizon 2020 Research and Innovation Programme under the Marie Skłodowska-Curie Grant Agreement 956150 (STORM-BOTS), No. 31221956150. H.Z. and A.P. are thankful to the financial support of the European Research Council (Starting Grant project ONLINE, No. 101076207, to H.Z. and Consolidator Grant project MULTIMODAL, No. 101045223, for A.P.).

## Author contributions

Z.D. discovered the phenomenon; H.Z. conceived the idea; Z.D. performed experiments under the supervision of H.Z. and A.P.; K.L. performed kinetic analysis. Z.D. wrote the first draft of the manuscript which was revised and finalized with help from others; All authors discussed and contributed to the project.

## Competing interests

The authors declare no competing interests.
