## [Transparent Peer Review file · Communications Materials]

A Light-Fueled Self-Oscillator That Senses Force

Corresponding Author: Professor Hao Zeng

Version 0:

Decision Letter:

Dear Professor Zeng,

Thank you for submitting your manuscript, "A Light-Fueled Self-Oscillator That Senses Force", to Communications Materials. It has now been seen by 2 referees, whose comments are appended below. You will see that while they find your work of interest, some important points are raised. We are interested in the possibility of publishing your study in Communications Materials, but would like to consider your response to these concerns in the form of a revised manuscript before we make a decision on publication.

In particular, Reviewer 2 asks how the oscillation curve can be used for certain analysis. Additionally, both Reviewers ask for further clarification on your observations.

We therefore invite you to revise and resubmit your manuscript, taking into account the points raised.

When submitting your revised manuscript, please include the following:

-A response letter with a point-by-point reply to each of the referee comments and a description of changes made. Please include the complete referee report in the response letter. Please note that the response letter must be separate to the cover letter to the editors.

-A marked-up version of the manuscript with all changes to the text in a different colored font. Please do not include tracked changes or comments. Please select the file type 'Revised Manuscript - Marked Up' when uploading the manuscript file to our online system.

-A clean version of the manuscript. Please select the file type 'Article File'.

-An updated <https://www.nature.com/documents/nr-editorial-policy-checklist.zip> Editorial Policy checklist, uploaded as a 'Related Manuscript File' type. This checklist is to ensure your paper complies with all relevant editorial policies. If needed, please revise your manuscript in response to these points. Please note that this form is a dynamic 'smart pdf' and must therefore be downloaded and completed in Adobe Reader. Clicking this link will download a zip file containing the pdf.

In the event that your manuscript is accepted we will provide detailed guidance on our journal policies and formatting. You may however wish to ensure that the manuscript complies with our house style at this stage. See our style and formatting guide (<https://www.nature.com/documents/commsj-phys-style-formatting-guide-accept.pdf>) and checklist (<https://www.nature.com/documents/commsj-phys-style-formatting-checklist-article.pdf>) for reference.

Data availability statements and data citations policy: All Communications Materials manuscripts must include a section titled "Data Availability" at the end of the Methods section or main text (if no Methods). More information on this policy, and a list of examples, is available at <http://www.nature.com/authors/policies/data/data-availability-statements-data->

[citations.pdf](http://www.nature.com/authors/policies/data/data-availability-statements-data-citations.pdf)><http://www.nature.com/authors/policies/data/data-availability-statements-data-citations.pdf>.

- Accession codes for deposited data
- Other unique identifiers (such as DOIs and hyperlinks for any other datasets)
- At a minimum, a statement confirming that all relevant data are available from the authors
- If applicable, a statement regarding data available with restrictions
- If a dataset has a Digital Object Identifier (DOI) as its unique identifier, we strongly encourage including this in the Reference list and citing the dataset in the Data Availability Statement.

DATA SOURCES: We strongly encourage authors to deposit all new data associated with the paper in a persistent repository where they can be freely and enduringly accessed. We recommend submitting the data to discipline-specific, community-recognized repositories, where possible and a list of recommended repositories is provided at <http://www.nature.com/sdata/policies/repositories>.

If a community resource is unavailable, data can be submitted to generalist repositories such as [figshare](https://figshare.com/) or [Dryad Digital Repository](http://datadryad.org/). Please provide a unique identifier for the data (for example a DOI or a permanent URL) in the data availability statement, if possible. If the repository does not provide identifiers, we encourage authors to supply the search terms that will return the data. For data that have been obtained from publically available sources, please provide a URL and the specific data product name in the data availability statement. Data with a DOI should be further cited in the methods reference section.

Please use the following link to submit your documents:

Link Redacted

We hope to receive your revised paper within three months; please let us know if you aren't able to submit it within this time so that we can discuss how best to proceed. If we don't hear from you, and the revision process takes significantly longer, we will close your file. In this event, we will still be happy to reconsider your paper at a later date, as long as nothing similar has been accepted for publication at Communications Materials or published elsewhere in the meantime.

Please do not hesitate to contact me if you have any questions or would like to discuss these revisions further. We look forward to seeing the revised manuscript and thank you for the opportunity to review your work.

Best regards,

Dr Jet-Sing Lee
Senior Editor
Communications Materials
orcid.org/0000-0002-6740-8700

Reviewers' comments:

Reviewer #1 (Remarks to the Author):

In this manuscript, the authors proposed a strategy to bypass the limitation in conventional sustained oscillation systems that rely on self-shadowing and demonstrated sustained oscillations without complete shadowing through applying an external mechanical force. Furthermore, they prove the generalization of the force-assisted self-oscillation principle across diverse deformation modes. Overall, this is interesting work. Therefore, I recommend its publication after the following detailed concerns have been well addressed.

1. In the manuscript, the authors demonstrate that the LCN strip was connected with a weight attached via a long, thin copper. The connection between the thread and the LCN strip is a spot, respective to the edge of the LCN strip, and the stress, from the weight, would not be uniformly distributed on the LCN strip. Will the phenomenon still be exhibited when the area of the connecting part increases? For example, how about connecting the same weight through two connecting points? Or the stress from the weight uniformly distributed on the whole area of the LCN strip edge?

2. When a weight is attached to the LCN strip, will it cause any changes to the modulus or the alignment of the LCN strip? It seems that when a weight is attached, the central part of the LCN strip, along the propagation of the thread, is constrained during deformations and the self-oscillation arises from the unsymmetric photomechanical deformations.

3. In Figure S7k, the oscillation under 167 mW cm⁻² was not symmetric, could the authors explain the reason for that?

Besides, considering the phenomenon arises as interplay between active photomechanical forces and stretching-induced resistance, when it comes to the off-axis sample in Figure S11, why does the 45° sample have larger rotating angles than the 30° sample? Will the 30° sample have larger deformation extent along the vertical axis when hung?

4. In supplementary video 2, the LCN strip with a 1.2g weight seems to keep rotating counterclockwise, could the authors explain for that? Will the weight and the thread rotate (deform) during the self-oscillating process? Will that affect the self-oscillating process?

Reviewer #2 (Remarks to the Author):

The authors report an LCN oscillator utilizing twisting-induced reductions in light absorption under constant light, exhibiting large oscillation amplitudes of up to 150°. This new oscillation mechanism is interesting and the results are solid and comprehensive. However, the concept of a force sensor is not quite clear, and I think there is a better way to show the potential of such a unique self-oscillator. Generally, I do like this idea, and I recommend it for publication after the authors address the following comments.

1. For the force sensor, I can only see the different oscillation performances (amplitude and frequency) under different loads. However, if the authors intend to emphasize its sensing capability or justify calling it a "sensor," they should demonstrate how the oscillation curve can be used to analyze an unknown load and accurately get its weight based on oscillation parameters. Providing a quantitative method or calibration curve linking oscillation characteristics to weight measurements, or giving a demo to show that you can get the unknown weight by analyzing the oscillation curve would strengthen the claim that this system functions as a true force sensor.

2. As for Fig. 4j, why do all three kinds of LCN show similar frequencies when the load exceeds 1 g? Does it mean that the alignment of the LCN plays a minimal role at higher load influencing the output, such as power density, in such a self-oscillation?

3. Compared to the bending-mode self-oscillated LCN with a frequency of over 20 Hz, why does such a rotational mode show a much lower frequency of around 1 Hz? Thus, I wonder if the output of such oscillation modes is significantly lower than those of bending-mode oscillators. So, what is the unique application area for this mode since it seems not quite suitable as a soft actuator for now? And if the goal is simply load measurement, why not just use a conventional weighing scale, even with higher resolution and simpler implementation?

4. The authors claim that the twisting oscillation reaches a large oscillation of 150°, much larger than the traditional bending mode. However, this might not be a fair comparison, due to the fact that the power output in the twisting mode is probably similar to, if not smaller than, that in the bending mode. The authors might also review other works of LCEs with twisting motion, some of which achieve oscillation or twisting angle close to 150° (<https://doi.org/10.1038/s41467-021-23562-6>, <https://doi.org/10.1002/adma.202107840>, <https://doi.org/10.1002/adma.202401140>, etc).

Communications Materials is committed to improving transparency in authorship. As part of our efforts in this direction, we are now requesting that all authors identified as 'corresponding author' create and link their Open Researcher and Contributor Identifier (ORCID) with their account on the Manuscript Tracking System prior to acceptance. ORCID helps the scientific community achieve unambiguous attribution of all scholarly contributions. You can create and link your ORCID from the home page of the Manuscript Tracking System by clicking on 'Modify my Springer Nature account' and following the instructions in the link below. Please also inform all co-authors that they can add their ORCIDs to their accounts and that they must do so prior to acceptance.

Version 1:

Decision Letter:

Dear Professor Zeng,

Your manuscript titled "A Light-Fueled Self-Oscillator That Senses Force" has now been seen again by our referees, whose comments appear below. In light of their advice I am delighted to say that we are happy, in principle, to publish a suitably

revised version in Communications Materials.

We therefore invite you to edit your manuscript to comply with our journal policies and formatting style in order to maximise the accessibility and therefore the impact of your work.

EDITORIAL REQUESTS

* Your manuscript should comply with our policies and format requirements, detailed in our style and formatting guide (<https://www.nature.com/documents/commsj-phys-style-formatting-guide-accept.pdf>).

* Please edit your manuscript according to the editorial requests in the attached table, and outline revisions made in the right hand column. If you have any questions or concerns about any of our requests, please do not hesitate to contact me. It is important that each request be addressed in order to avoid delays in accepting your manuscript. Please upload the completed table with your manuscript files as a Related Manuscript file.

* The editorial requests table also includes a full list of the files that must be provided upon resubmission. Please upload your files according to this table.

OPEN ACCESS

Communications Materials is a fully open access journal. Articles are made freely accessible on publication. For further information about article processing charges, open access funding, and advice and support from Nature Research, please visit <https://www.nature.com/commsmat/open-access>

Please use the following link to submit your revised files:

Link Redacted

We hope to hear from you within two weeks; please let us know if the process may take longer.

Best regards,

Dr Jet-Sing Lee
Senior Editor
Communications Materials
orcid.org/0000-0002-6740-8700

REVIEWERS' COMMENTS:

Reviewer #1 (Remarks to the Author):

The authors have addressed the review concerns adequately. I am therefore happy to recommend its acceptance as is.

Reviewer #2 (Remarks to the Author):

The authors have addressed all of my concerns, and I have no further comments.

Re: A Light-Fueled Self-Oscillator That Senses Force (Research Article, COMMSMAT-25-0198) by Z. Deng, et al.

Responses to the Reviewers

Reviewer #1: *In this manuscript, the authors proposed a strategy to bypass the limitation in conventional sustained oscillation systems that rely on self-shadowing and demonstrated sustained oscillations without complete shadowing through applying an external mechanical force. Furthermore, they prove the generalization of the force-assisted self-oscillation principle across diverse deformation modes. Overall, this is interesting work. Therefore, I recommend its publication after the following detailed concerns have been well addressed.*

Our answer: We are grateful for the reviewer's positive comments.

1. In the manuscript, the authors demonstrate that the LCN strip was connected with a weight attached via a long, thin copper. The connection between the thread and the LCN strip is a spot, respective to the edge of the LCN strip, and the stress, from the weight, would not be uniformly distributed on the LCN strip. Will the phenomenon still be exhibited when the area of the connecting part increases? For example, how about connecting the same weight through two connecting points? Or the stress from the weight uniformly distributed on the whole area of the LCN strip edge?

Our answer: We thank the reviewer for raising this very important point. To invest the influence of connecting points on the self-oscillation behaviour, we conducted additional experiments by attaching the same mass (2.4 g) to the strip via 1, 2, or 5 identical copper wires.

As shown in additional Supplementary Fig. 7, across all illumination intensities tested, increasing the number of connection points from one to five alters the peak rotation angle slightly by less than 10% and the oscillation frequency within 5%. We have added discussions to the revised manuscript to better clarification. Now, it reads,

When varying the number of connecting points by increasing the number of wires, the oscillation dynamics remain consistent under an identical external load. Only minor variations are observed in the oscillation amplitude (<10%) and frequency (<5%) when transitioning from a single-point to a five-point contact (Supplementary Fig. 7). These small differences indicate that our self-oscillation mechanism is highly robust to changes in contact area and largely insensitive to variations in stress distribution at the strip's end.

Supplementary Figure 7. Multi-thread connection. a) Oscillation amplitude and b) frequency as a function of the number of connection via 1, 2, 5 wires under three light intensities. The error bars are displayed as mean values \pm standard deviation ($n = 3$). The same sample was measured repeatedly. Relative variation in (c) amplitude and (d) frequency, normalized to the single-point contact case. A_1 and f_1 denote the amplitude and frequency for the single-point connection, respectively. Load condition: 2.4 g weight. Irradiation wavelength: 532 nm.

2. When a weight is attached to the LCN strip, will it cause any changes to the modulus or the alignment of the LCN strip? It seems that when a weight is attached, the central part of the LCN strip, along the propagation of the thread, is constrained during deformations and the self-oscillation arises from the unsymmetric photomechanical deformations.

Our answer: We thank the reviewer for this insightful comment. Our planar-aligned, highly crosslinked LCN possess a Young's modulus of about 20 MPa, and all applied loads in our experiments (up to a 2.4 g mass) remain well within the material's linear elastic regime and should not affect the material's stimuli-induced deformability. To confirm this experimentally, we performed two complementary tests (Supplementary Fig. S10):

1) Young's modulus before and after static loading: Planar-aligned LCN strips subjected to a 2.4 g load for 4 hours, display similar stress-strain curves, indicating no permanent stiffening or softening.

2) Repeated loading–unloading cycles: When the same LCN strips under cyclic loading (5 consecutive cycles of attaching/removing a 2.4 g weight), oscillation frequencies vary by <5%

across light intensities (200–233 mW cm^{-2}), demonstrating robust reversibility without cumulative effects (Supplementary Fig. 10c).

We have added an additional discussion into the revised text, now it reads,

It is worth noting that while the applied load affects the oscillatory behavior, it has a negligible impact on the material's intrinsic properties, such as Young's modulus (Supplementary Fig. 10a–b). The crosslinked polymer strips exhibit reliable reversibility and robust oscillation dynamics under repeated loading-unloading cycles (Supplementary Fig. 10c).

Supplementary Figure 10. Negligible effect of stretching on material properties. a) Mechanical testing of LCN strips stretched along the alignment direction before and after 4 hours of loading. b) Calculated Young's modulus before and after loading. The error bars are displayed as mean values \pm standard deviation ($n = 3$). c) Oscillation frequency under repeated loading–unloading cycles at three light intensities. N denotes the number of loading cycles. The error bars are displayed as mean values \pm standard deviation ($n = 3$). The same sample was measured repeatedly. Load condition: 2.4 g weight. Irradiation wavelength: 532 nm.

We would also like to note that, although the threaded connection imposes a local constraint on the strip, it does not alter the bulk molecular alignment, material modulus, or the fundamental oscillation dynamics. Crucially, self-oscillation originates from a dynamic balance between the photomechanical driving force and the restoring torque generated by the suspended mass, rather than from localized constraints or uneven stress distributions.

3. In Figure S7k, the oscillation under 167 mW cm^{-2} was not symmetric, could the authors explain the reason for that? Besides, considering the phenomenon arises as interplay between active photomechanical forces and stretching-induced resistance, when it comes to the off-axis sample in Figure S11, why does the 45° sample have larger rotating angles than the 30° sample? Will the 30° sample have larger deformation extent along the vertical axis when hung?

Our answer: We thank the reviewer for these thoughtful observations.

The asymmetry in Fig. S7k can be interpreted via threshold proximity. Under illumination, the system operates near the threshold required to initiate and sustain oscillation under the 300 mg load. Near the threshold, the photomechanical force is only marginally strong enough to overcome the restoring force, making the dynamics highly sensitive to small imperfections from the material, light field, or slight disturbances from the ambient. These minor factors randomly affect the process and induce an asymmetrical oscillation pattern. With the applied

load increasing, the actuation becomes stronger and more deterministic, in turn, the oscillation profiles tend to become more symmetric and repeatable.

The 45° sample indeed shows a larger rotation amplitude than the 30° sample. This difference primarily arises from two factors:

i) Illumination intensity: The 45° sample was actuated under a higher light intensity (up to 500 mW cm⁻²) than the 30° sample (up to 300 mW cm⁻²), which leads to stronger photomechanical deformation and consequently a larger rotational amplitude.

ii) Deformation modes: While the 30° sample likely experiences more contraction along the vertical (gravity-aligned) axis due to its orientation, the 45° sample undergoes a more pronounced twisting deformation, which more effectively translates into a rotation motion in the experimental setting. Of note, 30° sample broke when driven above ~300 mW cm⁻², preventing us from directly matching the 500 mW cm⁻² condition used for 45°.

4. In supplementary video 2, the LCN strip with a 1.2g weight seems to keep rotating counterclockwise, could the authors explain for that? Will the weight and the thread rotate (deform) during the self-oscillating process? Will that affect the self-oscillating process?

Our answer: We appreciate the reviewer's careful reading and providing the insightful comment. Upon revisiting our raw data and high-speed video for the 1.2 g loading condition, we confirm that the strip indeed undergoes bidirectional oscillation, rotating clockwise in one half-cycle and counterclockwise in the next. To clarify this point, we present below a sequence of frames from the oscillation recording, where the alternating orientation of the arrow marker clearly illustrates the back-and-forth rotational motion.

During self-oscillation, the weight–thread assembly moves in unison with the LCN strip in the vertical direction, while the thread head and the LCN strip synchronize in rotational motion. The added weight at the bottom does not rotate. However, unexpected wind disturbances may cause some rotational motion or displacement of the thread and load. In our experiment, we used a 1.1-meter-long thread (50 times the length of the LCN strip) to minimize the potential influence of any unintended movement of the attached weight.

Reviewer #2: The authors report an LCN oscillator utilizing twisting-induced reductions in light absorption under constant light, exhibiting large oscillation amplitudes of up to 150°. This new oscillation mechanism is interesting and the results are solid and comprehensive. However, the concept of a force sensor is not quite clear, and I think there is a better way to show the potential of such a unique self-oscillator. Generally, I do like this idea, and I recommend it for publication after the authors address the following comments.

Our answer: We thank the reviewer for the comments. We appreciate the constructive feedback on how we could better clarify and strengthen the presentation of the force-sensing concept and the broader potential of the system.

1. For the force sensor, I can only see the different oscillation performances (amplitude and frequency) under different loads. However, if the authors intend to emphasize its sensing capability or justify calling it a "sensor," they should demonstrate how the oscillation curve can be used to analyze an unknown load and accurately get its weight based on oscillation parameters. Providing a quantitative method or calibration curve linking oscillation characteristics to weight measurements, or giving a demo to show that you can get the unknown weight by analyzing the oscillation curve would strengthen the claim that this system functions as a true force sensor.

Our answer: We thank the reviewer for this valuable suggestion, which helped us further clarify and substantiate the sensing capability of our self-oscillating system. To address this point, we performed a set of experiments demonstrating how oscillation characteristics (especially, frequency) can quantitatively determine unknown loads. The procedure involved two key steps:

i) Calibration: We systematically measured oscillation frequencies across a range of known masses (0.3–2.4 g) at fixed light intensity (200 mW cm⁻²), establishing a calibration curve that relates frequency to mass (Supplementary Fig. 17a).

ii) Blind validation: Five ‘unknown’ weights (0.5 g, 0.7 g, 0.9 g, 1.1 g, and 2.4 g) were then tested using the same setup. Their oscillation frequencies closely match values predicted from the calibration curve, with all relative errors below 8% (Supplementary Fig. 17b).

These results confirm that our system’s oscillatory response enables quantitative measurement of unknown loads, supporting its functionality as a light-fueled soft force sensor. We have added these results to Supplementary Fig. 17, along with a brief discussion to the revised manuscript to clarify the sensing mechanism.

The force-sensing capability of the system is further validated through a quantitative demonstration: by using oscillation frequency as the readout parameter, unknown applied forces can be reliably inferred from a calibration curve. As shown in Supplementary Fig. 17, the predicted weights exhibit strong agreement with the actual values, with relative errors below 8%. This result marks an important first step toward developing an active, light-driven soft force sensor.

Supplementary Figure 17. Estimation of unknown loads. a) Calibration curve obtained by fitting measured oscillation frequencies for known masses. b) Blind test for unknown weights and their corresponding relative errors. $Relative\ error = \frac{|Estimated - Actual\ mass|}{Actual\ mass}$. The error bars are displayed as mean values \pm standard deviation ($n = 3$). The same sample was measured repeatedly. Irradiation conditions: 532 nm, 200 $mW\ cm^{-2}$.

2. As for Fig. 4j, why do all three kinds of LCN show similar frequencies when the load exceeds 1 g? Does it mean that the alignment of the LCN plays a minimal role at higher load influencing the output, such as power density, in such a self-oscillation?

Our answer: We thank the reviewer for this insightful question. Indeed, in Figure 4j the oscillation frequencies of the planar, splay, and twist modes converge at loads above ~ 1 g. What we believe is that, under heavy loading, the load-induced restoring torque dominates the cycle time, effectively overshadowing differences in photomechanical response among the alignment modes.

However, alignment remains a critical factor in determining both the oscillation threshold and the oscillation amplitude, which together influence the mechanical power output. For example, at a 45° off-axis angle under a 2.4 g load: i) Planar samples require a light intensity exceeding $400\ mW\ cm^{-2}$ to initiate oscillation, ii) Splay samples begin oscillating above $167\ mW\ cm^{-2}$, and iii) Twist samples oscillate only above $283\ mW\ cm^{-2}$.

As illustrated in Figures 4c, 4f, and 4i, despite exhibiting similar oscillation frequencies at high loads, each alignment mode displays a distinct rotational amplitude at a given intensity, highlighting the role of molecular alignment in photomechanical oscillation.

We have added the following text to the revised manuscript:

Meanwhile, the oscillation frequencies of the planar, splay, and twist modes converge at loads above approximately 1 g. This convergence is likely due to the dominance of load-induced restoring torque under heavy loading, which governs the cycle time and effectively masks differences in photomechanical responses among the alignment modes.

3. Compared to the bending-mode self-oscillated LCN with a frequency of over 20 Hz, why does such a rotational mode show a much lower frequency of around 1 Hz? Thus, I wonder if the output of such oscillation modes is significantly lower than those of bending-mode

oscillators. So, what is the unique application area for this mode since it seems not quite suitable as a soft actuator for now? And if the goal is simply load measurement, why not just use a conventional weighing scale, even with higher resolution and simpler implementation?

Our answer: We thank the reviewer for these thoughtful comments, which give us the opportunity to clarify both the motivation behind our work and the potential applications of the force-assisted self-oscillator we present.

i) On the comparison with bending-mode LCN self-oscillators:

While it is true that bending-mode LCN oscillators can achieve frequencies above 20 Hz, we would like to emphasize that the goal of our work is not to compete on frequency or output power, but rather to expand the scope of self-sustained motion mechanisms in synthetic materials beyond self-shadowing-dominated designs.

Traditional high-frequency LCN oscillators rely heavily on precisely localized illumination, often using a laser to ensure effective self-shadowing. This imposes significant constraints on the illumination geometry and material deformation modes. In contrast, our system does not require shadowing to achieve oscillation, allowing for uniform, large-area illumination and a wider range of alignment configurations.

Moreover, bending-mode oscillators operate near their mechanical resonance, which fixes their frequency once initiated and limits dynamic tunability. In our system, the restoring force is externally tunable (e.g., by varying the applied load), allowing direct modulation of both oscillation frequency and amplitude in real time. This results in a non-resonant, light-tuned and force-field-assisted oscillatory mechanism, which is not accessible in conventional self-shadowing systems.

We also note that while bending-mode oscillators may achieve higher frequencies, they typically exhibit small displacements, limiting their usable mechanical stroke. In contrast, our system demonstrates large rotational amplitudes (up to 150°), contributing meaningfully to mechanical power output, which may benefit applications where large angular displacement, rather than speed alone, is functionally desirable. Moreover, our oscillating system operates with a load-to-weight ratio typically greater than 500 (4 mg LCN, several grams loading), whereas bending oscillators based on the self-shadowing effect usually have a load-to-weight ratio below 10 (Adv. Funct. Mater. 2024, 34, 2310955).

In short, our work does not aim to replace resonant bending-mode actuators but rather to introduce a complementary, far-from-equilibrium oscillatory framework that enables new design strategies in soft, light-driven systems, particularly in contexts involving external mechanical fields or where sensing and adaptability are essential.

ii) On the value of this system for load measurement compared to conventional scales:

We fully acknowledge that conventional weighing scales offer superior resolution and simplicity for direct force measurements. However, our intent is not to compete with such devices on precision, but to demonstrate a fundamentally different, material-intrinsic sensing strategy that operates without electronics, computation, or feedback control.

Our system functions as a light-powered, soft force transducer, in which oscillation frequency and amplitude inherently encode the applied load. This behavior arises from the interplay between light-induced deformation and external mechanical loading, yielding a fully passive, analog mechanism for force readout. This approach is inspired by biological mechanosensation,

where soft structures (e.g., cilia, flagella) autonomously adapt their motion in response to environmental forces without digital processing.

To demonstrate practical feasibility, we implemented a calibration-based sensing protocol (detailed in our response to Reviewer 2, Comment 2): after generating a frequency vs. load calibration curve, we performed blind tests with unknown weights and achieved <8% relative error in force estimation. This serves as a concrete proof-of-concept for using this system as a light-fueled, self-sustained soft force sensor.

Looking forward, we envision optimizing the material response for operation under ambient light (e.g., sunlight), and developing strategies for real-time, contactless optical tracking of oscillation parameters, paving the way for integration into soft devices that can autonomously perceive and adapt to mechanical stimuli.

In summary, while this approach is not meant to replace existing sensors or actuators, it opens up new opportunities for embedding functionality into soft, adaptive materials, laying groundwork for future bioinspired, intelligent systems capable of operating far from equilibrium.

4. The authors claim that the twisting oscillation reaches a large oscillation of 150°, much larger than the traditional bending mode. However, this might not be a fair comparison, due to the fact that the power output in the twisting mode is probably similar to, if not smaller than, that in the bending mode. The authors might also review other works of LCEs with twisting motion, some of which achieve oscillation or twisting angle close to 150° (<https://doi.org/10.1038/s41467-021-23562-6>, <https://doi.org/10.1002/adma.202107840>, <https://doi.org/10.1002/adma.202401140>, etc).

Our answer: We thank the reviewer for this important comment, which gives us the opportunity to clarify the context and scope of our comparison, and to position our work among existing LCE twisting systems.

Our intention in highlighting the large rotational amplitude (~150°) is not to claim superior power output over traditional bending-mode self-oscillators, but rather to showcase a distinct deformation mode that complements the existing repertoire of photomechanical oscillators. In bending-mode LCN oscillators, oscillation amplitudes are often limited due to geometric constraints and localized illumination, while in our force-assisted configuration, the rotational degrees of freedom enable larger angular displacements even under uniform illumination.

We agree that power output is determined by both amplitude and frequency, and in this respect, the lower oscillation frequency (~1 Hz) in our system means that its mechanical power density is likely incomparable to that of higher-frequency oscillators. However, our approach provides several distinct advantages:

- i) **Dynamic tunability:** Both frequency and amplitude in our system can be continuously adjusted by varying the applied load, whereas resonant bending-mode frequencies are effectively fixed once oscillation begins.
- ii) **Intrinsic force interaction:** The oscillator inherently couples to external mechanical fields, enabling mechano-responsive dynamics and the demonstration of active force sensing.
- iii) **Simplified operation:** The system requires only constant, large-area light, eliminating the need for precise beam alignment and can be extended to diverse alignment configurations.

We thank the reviewer for drawing our attention to other LCE twisting systems. We have added all three works to our References in the revised manuscript, and we draw the following comparison to clarify our unique contributions:

- i) Lv *et al.* reported spontaneous torsion under constant light via self-winding-induced shadowing, which exhibits ~ 2.3 Hz in the rotational oscillation (comparable to our 2 Hz for a half-length strip), yet significantly smaller rotational amplitude, $\sim 50^\circ$.
- ii) The work from Yang *et al.* and White *et al.* are not based on self-oscillation. In other words, the large twisting degrees they have attained are based on external modulation of the external field (e.g., on/off cycles).
- iii) All three examples have yet demonstrated the force-sensing capability.

These distinctions underscore that our work introduces a light-fueled, large-amplitude twisting self-oscillator that both broadens the design space of active materials and lays the foundation for soft, autonomous force sensors. We hope this comparison clarifies our manuscript's unique contribution to the field.